environmental science, physiology, ecology

climate change, maternal effects, multiple stressors, offspring performance, salinity, temperature

**Author for correspondence:**
Gabriela Torres
e-mail: gabriela.torres@awi.de

# Maternal and cohort effects modulate offspring responses to multiple stressors

Gabriela Torres[1,2], David N. Thomas[2], Nia M. Whiteley[3], David Wilcockson[4] and Luis Giménez[1,2]

[1]Biologische Anstalt Helgoland, Alfred Wegener Institute, Helmholtz Centre for Polar and Marine Research, Helgoland, Germany
[2]School of Ocean Sciences, College of Environmental Sciences and Engineering, Bangor University, Menai Bridge, UK
[3]School of Natural Sciences, College of Environmental Sciences and Engineering, Bangor University, Bangor, UK
[4]Institute of Biological, Environmental and Rural Sciences, Aberystwyth University, Aberystwyth, UK

GT, 0000-0002-4064-0585

Current concerns about climate change have led to intensive research attempting to understand how climate-driven stressors affect the performance of organisms, in particular the offspring of many invertebrates and fishes. Although stressors are likely to act on several stages of the life cycle, little is known about their action across life phases, for instance how multiple stressors experienced simultaneously in the maternal environment can modulate the responses to the same stressors operating in the offspring environment. Here, we study how performance of offspring of a marine invertebrate (shore crab *Carcinus maenas*) changes in response to two stressors (temperature and salinity) experienced during embryogenesis in brooding mothers from different seasons. On average, offspring responses were antagonistic: high temperature mitigated the negative effects of low salinity on survival. However, the magnitude of the response was modulated by the temperature and salinity conditions experienced by egg-carrying mothers. Performance also varied among cohorts, perhaps reflecting genetic variation, and/or maternal conditions prior to embryogenesis. This study contributes towards the understanding of how anthropogenic modification of the maternal environment drives offspring performance in brooders.

## 1. Introduction

Current and future estimates of climate-related changes in the marine environment have emphasized the necessity to understand the importance of multiple-driver (or stressor) effects on organisms, populations, communities and ecosystems [1–4]. The main issue is that climate change results in multivariate modifications in marine habitats with environmental variables reaching values that are near, or beyond, normal levels of variation. In such cases, environmental drivers of biotic responses may become stressors because they elicit a stress response, which may manifest as reductions in performance of individuals (e.g. lower survival or prolonged developmental periods). Understanding the cumulative impact of multiple drivers is considered to be one of the most pressing research goals in environmental sciences [5].

We are beginning to appreciate that the effects of multiple drivers cannot be predicted from studies on single environmental variables owing to the frequent detection of interactive effects [6–8]. Such interactions can be antagonistic or synergistic [8–10], depending on whether the presence of a driver exacerbates or mitigates the effect of a second driver. While synergistic interaction means that the effects are larger than the sum of the effects elicited by each single environmental driver, antagonistic interaction refers to effects that are less than the sum. The latter suggests some capacity of organisms to tolerate environmental change. The prevalence of each interaction is poorly understood as some studies

**Figure 1.** Scenarios of maternal modulation of offspring performance. In an optimal maternal environment ($M_E$: optimal; left panel) larvae exhibit an antagonistic response (TMLS) whereby reductions in performance, resulting from low salinity ($L_S$), are mitigated at moderately high temperatures ($L_T$). A suboptimal maternal environment ($M_E$: suboptimal) either pre-empts (middle panel) larvae to exhibit TMLS (i.e. responses to salinity are independent of temperature) or induces (right panel) a synergistic response (high-temperature exacerbates the stressful effects of low salinity). (Online version in colour.)

have reported synergistic interactions [4,7], while others have reported additive effects or antagonistic interactions [11–13]. Moreover, interactive effects of multiple stressors appear to vary across taxa, developmental stages and trophic levels.

An area that deserves attention concerns scenarios where environmental conditions fluctuate over time [4]. For instance, in coastal-estuarine habitats, salinity and temperature, as well as other environmental drivers can vary, especially in regions of freshwater influence, where spatial patterns in salinity are driven by estuarine or river plumes [14]. Temperature and salinity can also covary with season [15–17]. In summer, coastal-estuarine waters of lower salinity are usually warmer than coastal shelf seawaters. Natural variations associated with tidal cycles or freshwater runoff, or the fact that many organisms migrate across coastal gradients, means that individuals will experience periods of lower salinity coinciding with higher temperatures. Moreover, the covariation between temperature and salinity, as experienced by organisms in the summer, may reverse in the winter (brackish waters often being cooler than seawater [16,18]) and may be weaker in spring/autumn or during long periods of rainfall owing to cooler allochthonous inputs of freshwater from land.

Environmental fluctuation encountered during the maternal-offspring transition (e.g. hatching and larval release) can be critical. In brooding species, offspring are often released into a new environment [19] that contrasts with the conditions experienced in the egg mass during embryogenesis. Offspring appear to be particularly sensitive to genetic or developmental malfunctions [20] and environmental change may trigger a number of adaptive or non-adaptive phenotypic responses, [21–23]. Responses occurring at such time are dominated by maternal effects i.e. effects of the maternal environment or phenotype on offspring phenotype and performance [24–26]. At the evolutionary scale, theory predicts that maternal effects evolve under sudden environmental shifts or changes consistent with those of climate change [27]; in addition, maternal effects are expected to evolve in seasonal environments [28] such as temperate estuaries. Maternal effects, driven by environmental change, may occur before fertilization (prezygotic effects [26], e.g. variation in allocation of reserves into eggs), during embryogenesis (postzygotic effects [29–32]) or after hatching and offspring release (post-natal maternal effects). Few studies, however, have managed to study these

complexities and assessed the relative importance of changes in multiple environmental drivers before or after fertilization in brooding marine species. Yet, such studies are needed in order to obtain a more realistic picture of how organisms will cope with climate-driven modifications of the natural habitat. If maternal effects modulate offspring responses, then responses obtained from studies ignoring such effects either over- or underestimate the offspring capacity to cope with climate change.

Here, we evaluate the importance of maternal effects in modifying responses to temperature and salinity in early larval stages of the estuarine-coastal crab *Carcinus maenas*. *Carcinus maenas* is an euryhaline crab that is endemic to northern Europe [33–35] but considered a global invader. Larvae of *C. maenas* exhibit an antagonistic response to low salinity and increased temperature ('thermal mitigation of low salinity stress': TMLS; figure 1), whereby negative effects of low salinity on survival and developmental time are mitigated at high temperatures [36]. The most likely underpinning mechanism is an increase in osmoregulatory capacity at higher temperatures: thus, TMLS may be a consequence of this physiological plasticity. The TMLS is found in other coastal species and it is relevant in the light of climate change in that (moderate) warming may favour expansion towards coastal areas characterized by moderately low salinities [36]. However, the same study also found that responses vary among larvae from different females [36], which may be driven by variability in the maternal environment, for instance by the temperature and salinity experienced by females and embryos. Theoretically, salinity and temperature may alter embryonic developmental processes and hence modify larval performance in many possible ways. For example, suboptimal conditions in the maternal habitat (where embryos develop) may weaken or pre-empt the development of antagonistic responses (figure 1, pre-emption) or induce synergistic stress responses (figure 1, induction). In both cases, the assessment of offspring responses to stressors, without considering such maternal effects, will over-estimate the capacity of offspring to cope with climate-driven change.

In order to establish which scenario from figure 1 prevails in the shore crab, we studied the role of the maternal postzygotic environment in modifying larval performance in response to temperature and salinity. Our approach was to examine the effects

of salinity and temperature experienced by embryos during brooding, on the survival and developmental time of resultant first stage larvae, which were also exposed to the same stressors in a factorial design. In addition, we performed the experiments with larvae from females producing eggs at different times of the year (autumn versus early summer), in order to determine if postzygotic responses are consistent or if they vary among cohorts. Differences in responses among larvae from different cohorts (but otherwise kept under similar temperature-salinity conditions over both the embryonic and larval phases) should be driven by genetic differences among broods or the influence of prezygotic maternal effects [23]. Ultimately, we were interested in obtaining a more general picture on how offspring response may be modulated by maternal effects and how such modulation may vary among cohorts of females.

## 2. Material and methods

### (a) Animal husbandry, larval rearing and experimental design

Berried females of *C. maenas* were collected in the Menai Strait (north Wales, UK) in autumn (October–November) and early summer (May–June) and transferred to marine aquaria in the School of Ocean Sciences, Bangor University (UK). On the day of collection, embryos were staged and females were distributed in the experimental treatments. Females carrying eggs at early stages of development (i.e. at the initiation of the formation of the embryo) were distributed at random into four treatments consisting of two temperatures (15°C and 18°C) and two salinities (diluted seawater: 25 practical salinity units (PSU) and seawater: 35 PSU; salinity is expressed in PSU, equivalent to ppt, following standard convention in oceanography). Those treatment combinations represent suboptimal (moderate osmotic and thermal stress) and optimal conditions (electronic supplementary material, figure S1); preliminary experiments, using females from the same population, revealed that hatching of viable larvae was still possible at 15°C and a salinity of 25 PSU. Females (carapace width: average = 50.1 ± s.d. = 8.7 mm) were randomly distributed among the treatments. We ran preliminary correlation analyses with female size as a covariate, but we did not find any relationship. Therefore, we did not consider female body size in the subsequent analyses.

Females with embryos at the earliest possible developmental stage were used to ensure that the embryos experienced different temperature and salinity combinations for a minimum of two weeks (electronic supplementary material, table S1) and that they were exposed at the time of the formation of the first larval stage (Zoea I). This applied to all females from the autumn cohort, as well as the majority of the early summer cohort. For the latter, we had to discard a large number of berried females because of a parasitic infection within the egg mass. This led to only one surviving female being allocated to a salinity of 35 PSU and temperature of 18°C. Three females carrying fully formed embryonic zoea were also included, as this treatment combination is the same as natural summer conditions. These females were exposed to the treatment for 4 days before hatching (electronic supplementary material, table S1). Electronic supplementary material, figure S2 shows that, for this group, there is little variation in survival among broods.

The resulting larvae from the brooding females were held in six different combinations of temperature and salinity (15, 18 and 24°C, and 20 and 35 PSU) in a fully factorial design, representing the offspring environment. The salinities were chosen to reflect the higher tolerance to lower salinity of larvae (20 PSU) compared to the embryos (25 PSU). In addition, we added a higher temperature to test effects of extreme temperatures on the larvae (24°C). In

total, this gave 24 different combinations of embryonic and larval temperature and salinity conditions (electronic supplementary material, figure S1). Each berried female occupied an individual aquarium (volume: 3 l) supplied with fully aerated seawater. Aquaria were placed in two holding tanks (1.5 m length × 1.0 m width × 0.5 m height) thermostatically controlled at the desired temperature (15 or 18°C, respectively) by heating/cooling units. Each aquarium was supplied with natural seawater or appropriately diluted seawater for low salinities (see below for details). Water was taken from the Menai Strait, which was filtered (0.2 μm), ultraviolet (UV)-treated and aerated prior to use (at a salinity of 34 PSU and temperature of 15°C, pH = 8.00, $A_T$ = 2286 μmol kg$^{-1}$, dissolved inorganic carbon = 2140 μmol kg$^{-1}$, $pCO_2$ = 599 μatm; N.M. Whiteley 2014, unpublished observations). Twice a week, and two hours prior to the water change, females were offered mussels as food. Larvae were held in 100 ml filtered, UV-treated, aerated seawater or appropriately diluted seawater for low salinities in open necked shallow beakers and placed within temperature-controlled incubators (LMS, series 4, UK). Twenty-four hours prior to each water change, seawater dilution (for the lower salinity treatments) was achieved in separate holding tanks using a conductivity metre (WTW 315i) to determine salinity in natural seawater mixed with appropriate quantities of de-chlorinated tap water. Both females and larvae were maintained with a photoperiod 12 L : 12 D (light : dark hours). During the embryonic and larval exposures, temperature and salinity were measured daily while the water was replaced. Readings were stable throughout the incubations (variation less than a salinity of 0.1 PSU or 0.1°C).

Larvae hatched from each female were assigned randomly to each of six treatments, in five replicates (10 freshly hatched Zoea I each), each one consisting of a 100 ml beaker; all females produced sufficient larvae for experiments (*C. maenas* fecundity ca 180.000 embryos per clutch [37]). Larval rearing followed standard methods [36,38]: seawater and food (*Artemia* sp. ad libitum: 5 individuals ml$^{-1}$) were changed daily and dead larvae were recorded and discarded. The experiment finished when all larvae died or moulted to Zoea II. We quantified larval performance as survival (i.e. the proportion of initial Zoea I reaching Zoea II) and the duration of development (i.e. the time of development from hatching until moult to Zoea II).

### (b) Data analysis

Larval performance was evaluated as survival and duration of development of the first zoeal stage. It is at this stage when maternal effects are likely to be more important; in addition, the TMLS is well developed during the first zoeal stage [36]. Survival data (proportion) were first adjusted using the equation $p' = [p(n-1)/n + 0.5]/n$, ($n = 10$ individuals) and then analysed after logistic (=logit) transformation [39], following Griffen *et al.* [8]. For survival, we applied a five-way factorial model containing embryonic salinity ($E_S$), embryonic temperature ($E_T$), larval salinity ($L_S$), larval temperature ($L_T$) and season ($S$). We use the term 'embryonic' instead of 'maternal' in order to emphasize our focus on postzygotic maternal effects. There was an additional factor, female ($F$) which represents the within-cohort variation in the responses; i.e. variation in responses of individuals originating from different females and experiencing the same environments as embryos and larvae. We did not separate the embryos from the mothers because embryonic development and hatching are impaired when embryos are isolated from the mother [40]. Thus, the factor female was nested in the interaction between embryonic temperature, embryonic salinity and season, because each female belonged to a season and its respective embryos experienced a specific salinity-temperature combination. The between-cohort effect is captured by the term ($S$) and represents differences in the responses among individuals originating from females

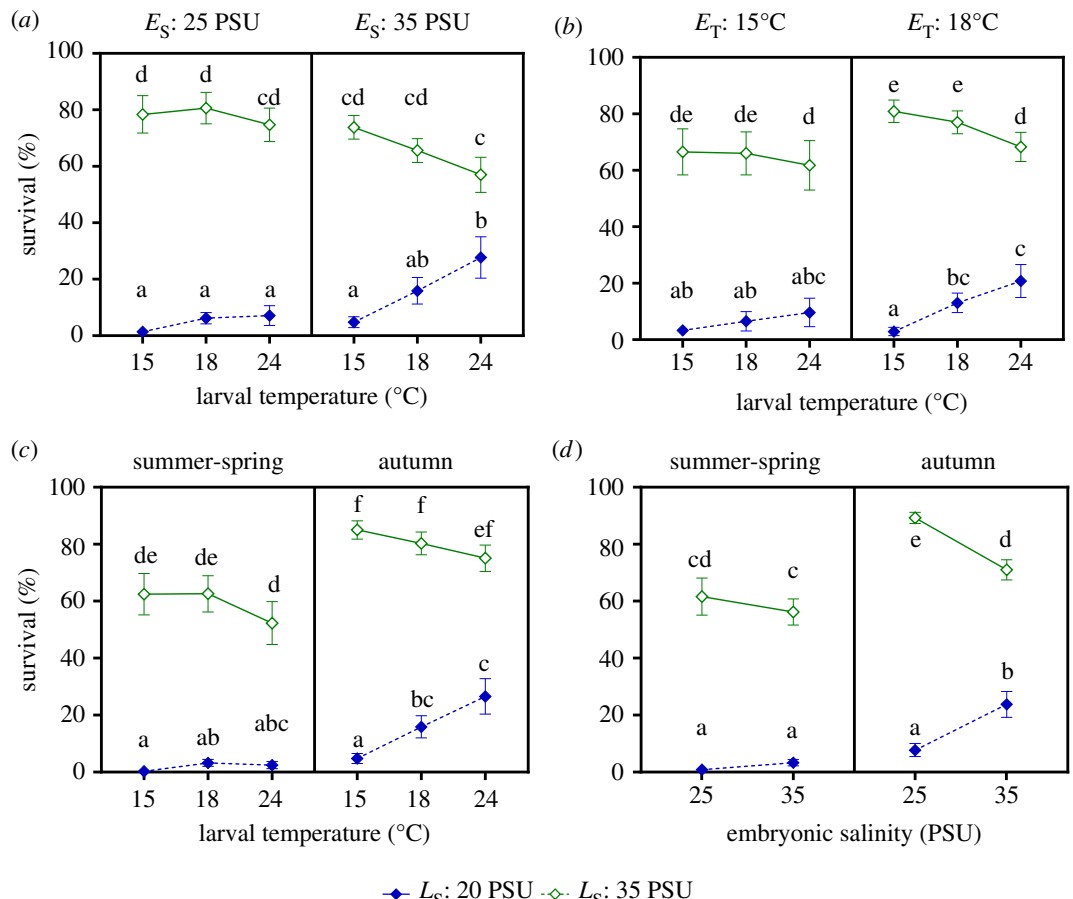

**Figure 2.** Survival of *Carcinus maenas* larvae to Zoea II. (*a*) Interaction between embryonic salinity ($E_S$), larval temperature ($L_T$) and larval salinity ($L_S$). (*b*) Interaction between embryonic temperature ($E_T$), larval temperature and larval salinity. (*c*) Interaction between season ($S$), larval temperature and larval salinity. (*d*) Interaction between season, embryonic salinity and larval salinity. Different letters indicate significant differences among the specific treatment combinations plotted within each panel. Values shown as mean ± standard error among larvae hatched from $n$ (see the electronic supplementary material, table S1) different females. (Online version in colour.)

belonging to different cohorts and experiencing the same environments as embryos and larvae.

The duration of development was analysed using the data corresponding to the larvae reared in seawater because we had high mortality rates at a lower salinity of 20 PSU (see Results). The starting model was reduced to a four-way factorial model (the factor 'larval salinity', $L_S$, was dropped), keeping female ($F$) as a random factor. This model still enabled us to test up to fourth-order interactive effects (e.g. $E_T : E_S : L_T : S$). Statistical analyses were run (separately) on the raw and log-transformed data in order to determine if interactive responses observed on the raw data (interaction term retained during model selection) reflected proportional effects (the same term is not retained for log-transformed data).

Statistical analyses were carried out through linear mixed model effects [41], in R [42] using the package nlme [43]. In addition to the terms in the model, we controlled for variance heterogeneity among replicates (using the VarIdent constructor function [43]). Although our design was fully replicated, our attempts at fitting the full model led to situations of a singular matrix, suggesting that some components were not estimated [44]. When this occurred, we followed procedures outlined by Bolker *et al.* [44] and reduced the complexity of the starting model. We used a combination of model selection (based on Akaike information criteria (AIC)) and hypothesis testing approaches as follows. First, model selection was applied through the backwards approach (i.e. starting with the full model) and then ranking models through AIC, detecting differences between the model with the lowest AIC and any other model (ΔAIC). When the simplest model had the lowest AIC, that model was selected; if ΔAIC was greater than 3, the model with lower AIC was selected

irrespective of differences in complexity. Hypothesis testing (likelihood ratio tests) was applied only when ΔAIC was less than 3, and the most complex model had the lower AIC. When models differed significantly ($p < 0.05$), the one with lower AIC was selected; in the opposite situation, the principle of parsimony was applied and the model with lower number of parameters was selected. Model selection was applied in two steps: (i) on the random structure (i.e. variance heterogeneity and effects of female of origin, interacting with larval salinity and temperature) using the restricted maximum-likelihood method (REML); then (ii) on the fixed structure (i.e. effects of season, embryonic and larval salinity and temperature) through maximum likelihood (ML).

## 3. Results

### (a) Survival to Zoea II

Larval survival showed complex responses to changes in temperature and salinity in the offspring environment (electronic supplementary material, table S2). Larval survival showed an antagonistic response, called thermal mitigation of low salinity stress (TMLS [36]). Survival was lower at low salinity, but improved at 18°C or 24°C, compared with survival at 15°C (figure 2*a*–*c*). Plots of larval survival per brooding female showed that survival at low salinity peaked either at 18°C, or at 24°C, depending on female (electronic supplementary material, figure S3). Consistent with the TMLS response, temperature and salinity interacted to influence survival which differed from the multiplicative model of survival. For

example, taking 15°C and a salinity of 35 PSU as the control conditions (average survival = 76%), the expected independent effect of each variable is the product of the survival probability observed at increased temperature, but optimal salinity (=66%), and that observed at reduced salinity, but optimal temperature (=3%). The expected effect under the hypothesis of independence is 2% (=0.66 × 0.03), which is eight times lower than observed survival (17%) at 24°C and salinity of 20 PSU and is consistent with the TMLS.

The TMLS response was modulated by embryonic salinity (figure 2a, electronic supplementary material, table S2 : $E_S : L_S : L_T$ : likelihood ratio (LR) = 15.21, $p < 0.001$) and embryonic temperature (figure 2b, electronic supplementary material, table S2 : $E_T : L_S : L_T$ : LR = 8.73, $p = 0.013$). The mitigation effect was weaker when embryos were kept at low salinity (=25 PSU). In larvae exposed to salinity of 20 PSU and 24°C, average survival was only approximately 7% when larvae hatched from embryos kept at salinity of 25 PSU, while average survival was approximately 28% when larvae hatched from embryos kept at a salinity of 35 PSU, which was a fourfold difference. In addition, the mitigation effect was weaker when embryos were kept at 15°C. For instance, larvae exposed to a salinity of 20 PSU and temperature of 24°C had a survival of approximately 10% when hatched from embryos kept at 15°C, but under the same larval conditions, survival was approximately 21% when embryos were previously kept at 18°C, marking a twofold increase in survival. In summary, low temperature (15°C) or low salinity (25 PSU) experienced at the embryonic stage weakened the thermal mitigation of low salinity stress.

The magnitude of TMLS varied between cohorts and among females of the same cohort. Larvae hatching from females of the autumn cohort showed stronger TMLS than those of the spring-summer cohort (figure 2c; electronic supplementary material, table S2 : $S : L_S : L_T$ : LR = 8.73, $p = 0.013$). When reared at a salinity of 20 PSU at 24°C, larvae from the spring-summer cohort showed an average survival of 2.4%, while those of the autumn cohort showed an average survival of 26% (i.e. 10-fold increase in survival). In addition, the larvae exposed to a salinity of 35 PSU from the autumn cohort had on average, higher survival than those of the spring-summer cohort (figure 2c). Within cohorts, female effects (retained in the random structure of the model) consisted mainly in variations in the strength of the TMLS (electronic supplementary material, figure S3a): this is shown as an important variation in survival at low salinity when larvae were exposed to 24°C, observed more clearly in the autumn cohort. At high salinity (electronic supplementary material, figure S3b), the female effect appears to occur irrespective of larval temperature conditions.

The cohorts also differed in terms of the sequential effects of embryonic salinity and larval salinity (figure 2d; electronic supplementary material, table S2 : $S : E_S : L_S$ : LR = 5.60, $p = 0.018$). For the autumn cohort, high embryonic salinity ameliorated the effect of low larval salinity on survival, but such an effect did not occur in the spring-summer cohort. For the autumn cohort, the survival of larvae hatching from embryos kept at a salinity of 25 PSU was approximately 8% while those hatching from embryos kept at a salinity of 35 PSU had a survival of approximately 24% (i.e. a threefold increase). There were also significant differences in survival between cohorts in response to embryonic salinity and temperature (electronic supplementary material, table S2 : $S : E_S : E_T$ : LR = 5.56, $p = 0.018$), but these effects were weak (electronic

supplementary material, figure S4a) and not detected by post-hoc tests. The same was true for the effect of the embryonic salinity and larval temperature on survival (electronic supplementary material, table S2 : $S : E_S : L_T$ : LR = 7.21, $p = 0.027$; electronic supplementary material, figure S4b) and in both cases survival was generally higher in larvae from the autumn cohort.

### (b) Development time to Zoea II

The duration of development had a complex fourth-order inter-active response when analyses were based on the raw data (figure 3; $S : E_S : E_T : L_T$ : LR = 7.89, $p = 0.02$; model selection summarized in the electronic supplementary material, table S3). Duration of development was driven mainly by larval temperature ($L_T$); as expected, larvae developed faster at higher temperatures with average differences of 2 to 4 days between larvae reared at 24°C and 15°C (effects of larval salinity, were not tested owing to high mortality at low salinity).

In the spring-summer cohort (figure 3a), low embryonic salinity and temperature induced a stress response in larvae reared at 15°C: development of such larvae was ca 2 days longer than when exposed to other embryonic conditions (at the same larval temperature). Such a stress response was also absent from the autumn cohort (figure 3b).

After logarithmic transformation, the four-way inter-action was dropped from the model but the interactive effect of embryonic salinity and temperature was retained ($S : E_S : E_T$ : LR = 9.59, $p = 0.002$; model selection summarized in electronic supplementary material, table S4). In both seasons, the effect of temperature on duration of development was stronger in the spring-summer cohort at low embryonic salinities. Duration of development was longer in the spring-summer than in the autumn cohort. Larval temperature had a strong effect which varied with cohorts ($S : L_T$ : LR = 10.16, $p = 0.006$): the duration of development at 24°C was about 70% of that at 15°C for the autumn cohort and 60% for the spring-summer cohort.

## 4. Discussion

We found that postzygotic maternal effects can modulate performance of offspring of the shore crab C. maenas in response to salinity and temperature, and that such responses vary among seasonal cohorts. The main response was observed in terms of survival, where we found further evidence for a thermal mitigation of low salinity stress (TMLS [36]), now extended to our local population of the Irish Sea. Developmental duration showed a response, consistent with TMLS: duration was extended in larvae reared at low temperatures when such larvae hatched from embryos reared at low temperature and low salinity. Both responses are manifested when larvae are reared at low temperatures. At least the TMLS may be based on an increase in osmoregulatory capacity of the first zoeal stage [34] at high temperatures: osmoregulation is usually enhanced at high temperatures [45,46] as well as increases the capacity of mitochondria to produce ATP [47] and the ability to repair damage.

The TMLS response of the larvae was modulated by salinity and temperature experienced during embryogenesis. Reduced salinity and temperature did not fully pre-empt (figure 1) but weakened the capacity of the larvae to exhibit TMLS. The fact that a strong TMLS was observed after high embryonic

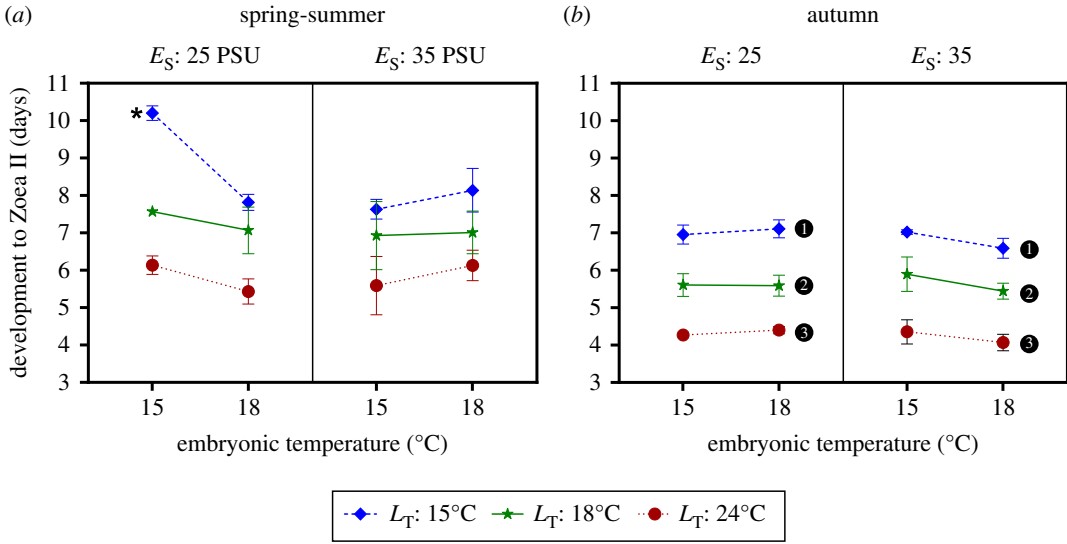

**Figure 3.** Development duration of *Carcinus maenas* larvae to Zoea II in seawater (i.e. $L_S = 35$). Four-way interaction between season (S), embryonic temperature ($E_T$) and salinity ($E_S$), and larval temperature ($L_T$). (a) Spring-summer cohort. (b) Autumn cohort. In (a), asterisk indicates significant differences among larvae exposed to 15°C. In (b) different numbers beside the symbols indicate significant differences between larval temperatures. Values shown as mean ± standard error among larvae hatched from $n$ (see the electronic supplementary material, table S1) different females. (Online version in colour.)

temperature may have important implications in that larvae will be more capable of surviving decreased salinities in a warming world owing to the combined action of temperature operating on both the embryonic and larval phases. Because TMLS is a common feature in some coastal crustaceans [48–50], an important question is whether moderate warming in the maternal habitat (at the time of embryogenesis) may further increase the magnitude of TMLS and favour the invasion of those near shore habitats characterized by moderately low salinities. Moderately increased temperature can lead to adaptive transgenerational plasticity, although extreme temperatures can disrupt adaptive plasticity [51].

An important question concerns the mechanisms underpinning the maternal modulation of larval responses. In estuarine species, there is a strong modulation of salinity tolerance through acclimatory responses [30,49], (i.e. a form of developmental plasticity whereby the larval tolerance to low salinity increases if embryos are also exposed to low salinity), based on an increase in osmoregulatory capacity [52]. However, we did not find evidence of embryonic acclimation to low salinities in *C. maenas*; by contrast, low salinity experienced during embryogenesis weakened the TMLS. Perhaps, exposure to low salinity depletes energy reserves during embryogenesis [53,54] with a resulting decrease in larval reserves, survival and developmental rate. By contrast, exposure of embryos to optimal temperatures can result in wider larval tolerances to temperature and salinity [55] which may accelerate the formation of osmoregulatory tissue.

How does the weakening of TMLS or the induction of a stress response (observed as extended the duration of development) relate to the various known maternal effects [25]? Our results need to be interpreted in the context of ontogenetic shifts in physiological tolerance because preliminary experiments showed that ovigerous females incubated at a salinity of 20 PSU rejected their eggs (G. Torres 2008, unpublished observations), while larvae were able to tolerate this salinity. For this reason, we cannot establish correspondences with, for instance, the concept of adaptive matching, whereby the best offspring performance occurs when the maternal and offspring environments coincide [25]. Ontogenetic shifts in

physiological tolerance should be widespread in brooding species where embryogenesis takes place at habitat conditions that differ considerably from those experienced by larvae [30,38,49], or where embryogenesis and larval development take place over different seasons [56,57].

Another important result concerned the variation in the magnitude of TMLS and in the duration of development in larvae hatched from different cohorts (=broods produced in different seasons). Inter-cohort variation in response to climate-driven stressors is arising as a major feature and can appear at several time scales (e.g. bi-weekly [58]; seasonal [51]; this study; among years [36,58]). Inter-cohort variation in the performance of organisms is important because they can stabilize or de-stabilize population dynamics [59,60]. For *C. maenas*, we do not have sufficient information about the structure of populations and thus we can only speculate on how inter-cohort variation may affect the dynamics. Higher performance in larvae resulting from autumn embryos may contribute to recruitment by buffering offspring from potential suboptimal (winter-spring) conditions; this would be similar to a case where a few individuals of high quality secure resources and avoid a population crash [60]. Larvae resulting from autumn embryos may also have a disproportionate contribution to recruitment (as compared with larvae from the spring-summer cohorts) if conditions are optimal. Disproportionate contributions to the population biomass by embryos produced in winter (as compared to 'summer embryos') appear to occur in natural populations of another coastal crustacean (brown shrimp *Crangon crangon*) in the North Sea [61]. Such embryos hatch into larvae that have higher tolerance to food limitation than those of summer embryos [62], but the contribution of winter embryos to the population appears to occur through an additional number of factors (e.g. a seasonal pattern of mortality rate). In principle, inter-cohort variation may reflect genetic variation as well as pre- and postzygotic maternal effects; plasticity may arise because, for instance, the temperature experienced by parents and embryos in a summer-autumn cohort will be higher than that experienced by the spring cohort (e.g. see [51]). In our case, differences in performance among cohorts were detected

among individuals hatching from embryos kept at the same temperature and salinity. Given that the postzygotic conditions were kept constant, inter-cohort differences must reflect genetic variability or prezygotic effects. In addition, variation in the responses should also reflect longer term transgenerational plasticity, for instance grandparental effects [63], which can only be teased apart by experiments running over several generations.

Previous studies have pointed to the necessity to understand the role of within and transgenerational phenotypic plasticity and genetic variation [21,27,64] in determining the capacity of organisms to respond to climate change. By focusing on an invasive marine brooder, this work highlights the importance of postzygotic effects (see also [30,51,58]) as modulators of larval responses to multiple environmental drivers, which may be relevant to understand how brooders cope with climate change. Furthermore, this study highlights the need of cross-habitat conservation programmes in species undergoing habitat shifts, as conditions in the maternal habitat determine the provision for the offspring with the physiological machinery to tolerate environmental stressors in the larval habitat.

Ethics (use of invertebrates for experimental studies). The research presented in this paper complies with the guidelines from the directives 2010/63/ EU of the European parliament and of the Council of 22nd September 2010 on the protection of animals used for scientific purposes.

Data accessibility. The datasets used and/or analysed during the current study are available from the Dryad Digital Repository: https://doi. org/10.5061/dryad.47d7wm39g [65].

Authors' contributions. G.T. and L.G. conceived this study and designed the experimental methodology, analysed the data and led the writing of the manuscript. G.T. performed all experiments to collect the data. All authors discussed the results and contributed critically to the drafts and gave final approval for publication.

Competing interests. The authors declare no competing interests. The funding body did not influence the design of the study, collection, analysis and interpretation of data, nor the writing of the manuscript.

Funding. The research was supported by a Marie Curie Fellowship PEOPLE-IEF-2008 - award no. 235634 (MATE) awarded to G.T. G.T. acknowledges support by the Open Access Publication Funds of Alfred-Wegener-Institut Helmholtz-Zentrum für Polar-und Meeresforschung.

Acknowledgements. We are grateful to Katherine Griffith, Kathryn Mainwaring and Thomas Pape for their help maintaining adult crabs. We thank Berwyn Roberts (School of Ocean Sciences, Bangor University, UK) and Trevor Jones (Extramussel Limited, Refail Llanffinan, Llangefni, LL77 7SN, UK) for providing the berried females.

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
