## [Reviewer comments · Proceedings of the Royal Society B: Biological Sciences]

Review History

RSPB-2020-0492.R0 (Original submission)

Review form: Reviewer 1

Recommendation

Major revision is needed (please make suggestions in comments)

Scientific importance: Is the manuscript an original and important contribution to its field?

Good

General interest: Is the paper of sufficient general interest?

Good

Quality of the paper: Is the overall quality of the paper suitable?

Good

Is the length of the paper justified?

Yes

Should the paper be seen by a specialist statistical reviewer?

Yes

Do you have any concerns about statistical analyses in this paper? If so, please specify them explicitly in your report.

No

It is a condition of publication that authors make their supporting data, code and materials available - either as supplementary material or hosted in an external repository. Please rate, if applicable, the supporting data on the following criteria.

Is it accessible?

N/A

Is it clear?

N/A

Is it adequate?

N/A

Do you have any ethical concerns with this paper?

No

Comments to the Author

This study explores the effects of multiple stressors (temperature and salinity) experienced during embryogenesis in the maternal environment on the responses of the offspring to such stressors, in the invasive marine crab *Carcinus maenas*. The study is timely and relevant as it addresses two poorly explored issues in the context of climate change and biological invasions: 1) the interactive effects of multiple stressors and 2) transgenerational plasticity. The manuscript is well written. However, after a careful revision, I have detected an important aspect of the experimental design that needs clarification, some limitations that should be discussed as well as other minor issues that I list below. I hope my comments are useful to improve the manuscript.

Major issues

1. Maternal effects are defined as environmentally-driven (non-genetic) effects of the maternal environment on the traits of the offspring. Such effects have been typically studied on clonal organisms, as this minimizes variation arising from genetic differences. In this study, it is not clear how the experimental design allows discriminating between environmentally driven maternal effects and those associated to genetic variation. Authors state that by performing experiments with females from different seasons (summer vs autumn) they quantify variations in performance associated to genetic differences or prezygotic effects (L 129-132). So, if I understand well, they interpret the different responses found between individuals of the same season (intra-cohort) as maternal effects, and the differences between seasons (inter-cohort) as genetic differences and/or prezygotic effects. To my understanding, this is questionable because genetic variation is present both at the intra and inter-cohort level as long as you are comparing larvae from different females. This point needs to be clarified.

2. By exploring maternal effects from one generation to the next, potential pre-existing transgenerational effects are not controlled for. This is an unavoidable limitation because multiple generation studies are only feasible with species with short generation times, but it should be discussed.

Minor issues

Authors may consider the model of evolution of maternal effects proposed by Kuijper and Hoyle, 2015 *Evolution* 69-4: 950-968, on the evolutionary dynamics of maternal effects in highly variable or periodically fluctuating environments, to contextualize and interpret their results.

L 24 replace has with have

L 26 replace invertebrate with invertebrates

L 95-99 What physiological mechanisms underlie the mitigation effect of high temperature on low salinities? If there is information on this, please include it.

L 141-143 Were the females directly transferred to the treatments after field collection? Or were they allowed to habituate to laboratory conditions for some time before the experiments? If so, in which conditions were they maintained?

L 148-149 Why did you measure carapace width? Was it a covariate considered in your analyses?

L 161-163 Could you explain why the rearing conditions of the larvae are slightly different from that of the maternal environment? (i.e. different salinity in the low salinity treatment and one more temperature treatment for larvae)

L 168-178 In this section, authors should be clearer in explaining which procedures were used for each salinity treatment. For example, in line 168 "supply with seawater" refers only to the higher salinity treatment, but not to the lower salinity one, and the same in L 172-174. But from L 175, I guess authors refer to the low salinity treatment.

L 181-182 From each female, there were enough larvae to distribute them across 6 treatment combinations x 5 replicates x 10 individuals? Including information on the typical number of eggs laid by females of the study species would be useful

L 214-215 This part may not be clear for a reader non familiar with this analyses, authors should consider providing additional details.

L 363 In the context of this paper, it would be recommendable to specify that with "acclimatory responses" here you refer to a form of transgenerational or developmental plasticity, to distinguish it from within-generational acclimation responses.

Review form: Reviewer 2

Recommendation

Accept with minor revision (please list in comments)

Scientific importance: Is the manuscript an original and important contribution to its field?

Acceptable

General interest: Is the paper of sufficient general interest?

Good

Quality of the paper: Is the overall quality of the paper suitable?

Excellent

Is the length of the paper justified?

Yes

Should the paper be seen by a specialist statistical reviewer?

No

Do you have any concerns about statistical analyses in this paper? If so, please specify them explicitly in your report.

No

It is a condition of publication that authors make their supporting data, code and materials available - either as supplementary material or hosted in an external repository. Please rate, if applicable, the supporting data on the following criteria.

Is it accessible?

N/A

Is it clear?

N/A

Is it adequate?

N/A

Do you have any ethical concerns with this paper?

No

Comments to the Author

I have read the manuscript entitled "Maternal and cohort effects modulate offspring responses to multiple stressors" by Torres et al. which describes how post-zygotic maternal effects modulate offspring performance in the invasive crab *C. maenas* in response to salinity and temperature. Overall, I found the paper well-written and the discussion of observed embryonic responses well-presented and therefore have only a few, minor, grammatical and visual changes to suggest in my comments (see below).

Line 24: Either "Current concerns ... have" or "Current concern ... has"

Line 32: brooding instead of breeding?

Lines 37-39: Whereas the authors are correct that the number of studies of transgenerational plasticity (TGP) in brooders is small, the field is not entirely unexplored. See for example, Tanner RL, Bowie RCK, Stillman JH. 2020. Thermal exposure and transgenerational plasticity influence embryonic success in a bivalvline estuarine sea hare. *Mar Ecol Prog Ser* 634:199-211. <https://doi.org/10.3354/meps13207>

Cumbo VR. et al. 2013. Brooded coral larvae differ in their response to high temperature and elevated pCO₂ depending on the day of release. *Mar Biol* 160, 2903–2917 (2013).

<https://doi.org/10.1007/s00227-013-2280-y>

I therefore suggest the authors modify this sentence to note that their results support the data presented in these (and potentially other) studies. In addition, the authors should include discussion of this (and other relevant) literature in their paper where appropriate (e.g., in the Discussion).

Lines 48-49: missing parentheses surrounding e.g. clause. Suggest re-wording of "...changes in offspring performance in individuals" as it is oddly phrased.

Figure 1 first panel should read "M = Optimal" and "S = Suboptimal". Suggest combining the "Pre-emption" and "Induction" panels as the optimal condition (TMLS) data is currently being displayed twice. This would also help clarify that, under optimal conditions, the TMLS response is exhibited whereas, sub-optimal conditions can lead to one of two alternatives (pre-emption and induction).

Line 342: Change "were" to "where"

Line 365: Change "found" to "find"

Decision letter (RSPB-2020-0492.R0)

01-Apr-2020

Dear Dr Torres:

Your manuscript has now been peer reviewed and the reviews have been assessed by an Associate Editor. The reviewers' comments (not including confidential comments to the Editor) and the comments from the Associate Editor are included at the end of this email for your reference. As you will see, the reviewers and the Editors have raised some concerns with your manuscript and we would like to invite you to revise your manuscript to address them.

Research ethics:

Use of animals and field studies:

Please submit a copy of your revised paper within three weeks. If we do not hear from you within this time your manuscript will be rejected. If you are unable to meet this deadline please let us know as soon as possible, as we may be able to grant a short extension.

Best wishes,
Dr Robert Barton
mailto: proceedingsb@royalsociety.org

Associate Editor

Board Member: 1

Comments to Author:

Both reviewers were very positive about the manuscript but both also suggested some critical factors that the authors need to address. Chief among these issues is major concern 1 by reviewer 1. The authors need to be very clear in responding to this issue. If the effects observed cannot be clearly interpreted as maternal effects this would undermine the study considerably.

Reviewer(s)' Comments to Author:

Referee: 1

Comments to the Author(s)

This study explores the effects of multiple stressors (temperature and salinity) experienced during embryogenesis in the maternal environment on the responses of the offspring to such stressors, in the invasive marine crab *Carcinus maenas*. The study is timely and relevant as it

addresses two poorly explored issues in the context of climate change and biological invasions: 1) the interactive effects of multiple stressors and 2) transgenerational plasticity. The manuscript is well written. However, after a careful revision, I have detected an important aspect of the experimental design that needs clarification, some limitations that should be discussed as well as other minor issues that I list below. I hope my comments are useful to improve the manuscript.

Major issues

1. Maternal effects are defined as environmentally-driven (non-genetic) effects of the maternal environment on the traits of the offspring. Such effects have been typically studied on clonal organisms, as this minimizes variation arising from genetic differences. In this study, it is not clear how the experimental design allows discriminating between environmentally driven maternal effects and those associated to genetic variation. Authors state that by performing experiments with females from different seasons (summer vs autumn) they quantify variations in performance associated to genetic differences or prezygotic effects (L 129-132). So, if I understand well, they interpret the different responses found between individuals of the same season (intra-cohort) as maternal effects, and the differences between seasons (inter-cohort) as genetic differences and/or prezygotic effects. To my understanding, this is questionable because genetic variation is present both at the intra and inter-cohort level as long as you are comparing larvae from different females. This point needs to be clarified.

2. By exploring maternal effects from one generation to the next, potential pre-existing transgenerational effects are not controlled for. This is an unavoidable limitation because multiple generation studies are only feasible with species with short generation times, but it should be discussed.

Minor issues

Authors may consider the model of evolution of maternal effects proposed by Kuijper and Hoyle, 2015 *Evolution* 69-4: 950-968, on the evolutionary dynamics of maternal effects in highly variable or periodically fluctuating environments, to contextualize and interpret their results.

L 24 replace has with have

L 26 replace invertebrate with invertebrates

L 95-99 What physiological mechanisms underlie the mitigation effect of high temperature on low salinities? If there is information on this, please include it.

L 141-143 Were the females directly transferred to the treatments after field collection? Or were they allowed to habituate to laboratory conditions for some time before the experiments? If so, in which conditions were they maintained?

L 148-149 Why did you measure carapace width? Was it a covariate considered in your analyses?

L 161-163 Could you explain why the rearing conditions of the larvae are slightly different from that of the maternal environment? (i.e. different salinity in the low salinity treatment and one more temperature treatment for larvae)

L 168-178 In this section, authors should be clearer in explaining which procedures were used for each salinity treatment. For example, in line 168 "supply with seawater" refers only to the higher salinity treatment, but not to the lower salinity one, and the same in L 172-174. But from L 175, I guess authors refer to the low salinity treatment.

L 181-182 From each female, there were enough larvae to distribute them across 6 treatment combinations x 5 replicates x 10 individuals? Including information on the typical number of eggs laid by females of the study species would be useful

L 214-215 This part may not be clear for a reader non familiar with this analyses, authors should consider providing additional details.

L 363 In the context of this paper, it would be recommendable to specify that with “acclimatory responses” here you refer to a form of transgenerational or developmental plasticity, to distinguish it from within-generational acclimation responses.

Referee: 2

Comments to the Author(s)

I have read the manuscript entitled "Maternal and cohort effects modulate offspring responses to multiple stressors" by Torres et al. which describes how post-zygotic maternal effects modulate offspring performance in the invasive crab *C. maenas* in response to salinity and temperature. Overall, I found the paper well-written and the discussion of observed embryonic responses well-presented and therefore have only a few, minor, grammatical and visual changes to suggest in my comments (see below).

Line 24: Either “Current concerns ... have” or “Current concern ... has”

Line 32: brooding instead of breeding?

Lines 37-39: Whereas the authors are correct that the number of studies of transgenerational plasticity (TGP) in brooders is small, the field is not entirely unexplored. See for example, Tanner RL, Bowie RCK, Stillman JH. 2020. Thermal exposure and transgenerational plasticity influence embryonic success in a bivoltine estuarine sea hare. *Mar Ecol Prog Ser* 634:199-211. <https://doi.org/10.3354/meps13207>

Cumbo VR. et al. 2013. Brooded coral larvae differ in their response to high temperature and elevated pCO₂ depending on the day of release. *Mar Biol* 160, 2903–2917 (2013).

<https://doi.org/10.1007/s00227-013-2280-y>

I therefore suggest the authors modify this sentence to note that their results support the data presented in these (and potentially other) studies. In addition, the authors should include discussion of this (and other relevant) literature in their paper where appropriate (e.g., in the Discussion).

Lines 48-49: missing parentheses surrounding e.g. clause. Suggest re-wording of “...changes in offspring performance in individuals” as it is oddly phrased.

Figure 1 first panel should read “M = Optimal” and “S = Suboptimal”. Suggest combining the “Pre-emption” and “Induction” panels as the optimal condition (TMLS) data is currently being displayed twice. This would also help clarify that, under optimal conditions, the TMLS response is exhibited whereas, sub-optimal conditions can lead to one of two alternatives (pre-emption and induction).

Line 342: Change “were” to “where”

Line 365: Change “found” to “find”

Author's Response to Decision Letter for (RSPB-2020-0492.R0)

See Appendix A.

RSPB-2020-0492.R1 (Revision)

Review form: Reviewer 1

Recommendation

Accept as is

Scientific importance: Is the manuscript an original and important contribution to its field?

Good

General interest: Is the paper of sufficient general interest?

Good

Quality of the paper: Is the overall quality of the paper suitable?

Good

Is the length of the paper justified?

Yes

Should the paper be seen by a specialist statistical reviewer?

No

Do you have any concerns about statistical analyses in this paper? If so, please specify them explicitly in your report.

No

It is a condition of publication that authors make their supporting data, code and materials available - either as supplementary material or hosted in an external repository. Please rate, if applicable, the supporting data on the following criteria.

Is it accessible?

Yes

Is it clear?

Yes

Is it adequate?

Yes

Do you have any ethical concerns with this paper?

No

Comments to the Author

I am happy to see that authors have satisfactorily addressed and clarified all the issues pointed in the previous revision round. I congratulate them for the considerable improvement of the manuscript. I have no further concerns on this new version.

I have detected these typos:

L 89: Replace assess with assessed

L 448 (word document) or 376 (pdf file): "and genetic variation" is repeated

Review form: Reviewer 2

Recommendation

Accept with minor revision (please list in comments)

Scientific importance: Is the manuscript an original and important contribution to its field?

Good

General interest: Is the paper of sufficient general interest?

Good

Quality of the paper: Is the overall quality of the paper suitable?

Acceptable

Is the length of the paper justified?

Yes

Should the paper be seen by a specialist statistical reviewer?

No

Do you have any concerns about statistical analyses in this paper? If so, please specify them explicitly in your report.

No

It is a condition of publication that authors make their supporting data, code and materials available - either as supplementary material or hosted in an external repository. Please rate, if applicable, the supporting data on the following criteria.

Is it accessible?

Yes

Is it clear?

Yes

Is it adequate?

Yes

Do you have any ethical concerns with this paper?

No

Comments to the Author

The authors have adequately addressed my previous comments/suggestions and the revised manuscript is much improved. However, after careful revision, I feel that the Discussion is still slightly under-developed especially with regard to how this study fits within the context of transgenerational effects on population dynamics. It is currently unclear how variance in the observed physiological responses might affect population-level trajectories for this particular species. The authors do an excellent job of presenting possible effects of variation in general, either as a result of transgenerational or because of pre-zygotic effects, but more specific discussion of how this variation may affect populations of the study species *C. maenas* is needed. In addition, there are a few other, minor, grammatical issues that I list below that should be corrected before publication. I hope my comments are useful to improve the manuscript.

Line 25: Change to "climate-driven".

Line 25: Suggest to change "in particular on" to "particularly the".

Lines 29 - 33: Suggest simplifying the sentence to "Here, we study how performance of offspring of a marine invertebrate (shore crab *Carcinus maenas*) changes in response to two stressors

(temperature and salinity) experienced by brooding mothers during embryogenesis in different seasons."

Lines 43 – 44: Please include hyphens between words when they are used as a compound adjective (e.g., "multiple-driver (or stressor) effects").

Line 81 – 82: Change to "(i.e., effects of the maternal environment or phenotype on 82 offspring phenotype and performance)".

Line 118: Suggest to change to "... experienced by embryos during brooding on the survival ...".

Line 143: Change to "We ran ...".

Line 184: Check value for fecundity. Is it 180 or 180,000? Units?

Line 186: Remove "**".

Line 341: Remove "that" after "by contrast,".

Line 352: "... with, for instance, the concept ..."

Line 364: The authors should expand their discussion of how their data fit within the context of "stabilizing or destabilizing the population dynamics". Does the inter-cohort variance in this study have the potential to affect population dynamics of *C. maenas* at the sites of collection? If so, in what way(s)?

Decision letter (RSPB-2020-0492.R1)

20-May-2020

Dear Dr Torres

I am pleased to inform you that your manuscript RSPB-2020-0492.R1 entitled "Maternal and cohort effects modulate offspring responses to multiple stressors" has been accepted for publication in Proceedings B.

The referee(s) have recommended publication, but also suggest some minor revisions to your manuscript. Therefore, I invite you to respond to the referee(s)' comments and revise your manuscript. Because the schedule for publication is very tight, it is a condition of publication that you submit the revised version of your manuscript within 7 days. If you do not think you will be able to meet this date please let us know.

1) A text file of the manuscript (doc, txt, rtf or tex), including the references, tables (including captions) and figure captions. Please remove any tracked changes from the text before submission. PDF files are not an accepted format for the "Main Document".

2) A separate electronic file of each figure (tiff, EPS or print-quality PDF preferred). The format should be produced directly from original creation package, or original software format. PowerPoint files are not accepted.

3) Electronic supplementary material: this should be contained in a separate file and where possible, all ESM should be combined into a single file. All supplementary materials accompanying an accepted article will be treated as in their final form. They will be published alongside the paper on the journal website and posted on the online figshare repository. Files on figshare will be made available approximately one week before the accompanying article so that the supplementary material can be attributed a unique DOI.

Sincerely,

Dr Robert Barton

Associate Editor:

Board Member: 1

Comments to Author:

Thanks for your patience with the review process. We have now got feedback on the revisions from both original reviewers of your manuscript. Both were very positive about the careful revisions you provided, but both (particularly reviewer 2) had some minor issues they suggested you address before the manuscript can be accepted. Personally I have nothing else to add. Congrats on a great contribution to the literature. The world needs more work on the complexity of multiple stressor interactions, so keep it up!

Reviewer(s)' Comments to Author:

Referee: 1

Comments to the Author(s)

I am happy to see that authors have satisfactorily addressed and clarified all the issues pointed in the previous revision round. I congratulate them for the considerable improvement of the manuscript. I have no further concerns on this new version.

I have detected these typos:

L 89: Replace assess with assessed

L 448 (word document) or 376 (pdf file): "and genetic variation" is repeated

Referee: 2

Comments to the Author(s)

The authors have adequately addressed my previous comments/suggestions and the revised manuscript is much improved. However, after careful revision, I feel that the Discussion is still slightly under-developed especially with regard to how this study fits within the context of transgenerational effects on population dynamics. It is currently unclear how variance in the observed physiological responses might affect population-level trajectories for this particular species. The authors do an excellent job of presenting possible effects of variation in general, either as a result of transgenerational or because of pre-zygotic effects, but more specific discussion of how this variation may affect populations of the study species *C. maenas* is needed. In addition, there are a few other, minor, grammatical issues that I list below that should be corrected before publication. I hope my comments are useful to improve the manuscript.

Line 25: Change to "climate-driven".

Line 25: Suggest to change "in particular on" to "particularly the".

Lines 29 – 33: Suggest simplifying the sentence to "Here, we study how performance of offspring of a marine invertebrate (shore crab *Carcinus maenas*) changes in response to two stressors (temperature and salinity) experienced by brooding mothers during embryogenesis in different seasons."

Lines 43 – 44: Please include hyphens between words when they are used as a compound adjective (e.g., "multiple-driver (or stressor) effects").

Line 81 – 82: Change to "(i.e., effects of the maternal environment or phenotype on 82 offspring phenotype and performance)".

Line 118: Suggest to change to "... experienced by embryos during brooding on the survival ...".

Line 143: Change to "We ran ...".

Line 184: Check value for fecundity. Is it 180 or 180,000? Units?

Line 186: Remove "***".

Line 341: Remove "that" after "by contrast,".

Line 352: "... with, for instance, the concept ..."

Line 364: The authors should expand their discussion of how their data fit within the context of "stabilizing or destabilizing the population dynamics". Does the inter-cohort variance in this

study have the potential to affect population dynamics of *C. maenas* at the sites of collection? If so, in what way(s)?

Author's Response to Decision Letter for (RSPB-2020-0492.R1)

See Appendix B.

Decision letter (RSPB-2020-0492.R2)

27-May-2020

Dear Dr Torres

I am pleased to inform you that your manuscript entitled "Maternal and cohort effects modulate offspring responses to multiple stressors" has been accepted for publication in Proceedings B.

Open Access

Paper charges

You are allowed to post any version of your manuscript on a personal website, repository or preprint server. However, the work remains under media embargo and you should not discuss it

with the press until the date of publication. Please visit <https://royalsociety.org/journals/ethics-policies/media-embargo> for more information.

Sincerely,

Editor, Proceedings B
mailto:proceedingsb@royalsociety.org

Appendix A

RESPONSE TO REFEREES

We thank the reviewers for their helpful comments, which have helped us to improve the manuscript by clarifying our ideas and providing a better context for our study. Please find below, detailed comments on each issue and information on the changes made in the manuscript.

Please, find the revised MS + track changes at the end of the “Response to referees”.

Mail 01-Apr-2020

Comments to Author:

Both reviewers were very positive about the manuscript but both also suggested some critical factors that the authors need to address. Chief among these issues is major concern 1 by reviewer 1. The authors need to be very clear in responding to this issue. If the effects observed cannot be clearly interpreted as maternal effects this would undermine the study considerably.

Answer: please see detailed comments below.

Reviewer(s)' Comments to Author:

Referee: 1

Comments to the Author(s)

This study explores the effects of multiple stressors (temperature and salinity) experienced during embryogenesis in the maternal environment on the responses of the offspring to such stressors, in the invasive marine crab *Carcinus maenas*. The study is timely and relevant as it addresses two poorly explored issues in the context of climate change and biological invasions: 1) the interactive effects of multiple stressors and 2) transgenerational plasticity. The manuscript is well written. However, after a careful revision, I have detected an important aspect of the experimental design that needs clarification, some limitations that should be discussed as well as other minor issues that I list below. I hope my comments are useful to improve the manuscript.

Major issues

1. Maternal effects are defined as environmentally-driven (non-genetic) effects of the maternal environment on the traits of the offspring. Such effects have been typically studied on clonal organisms, as this minimizes variation arising from genetic differences. In this study, it is not clear how the experimental design allows discriminating between environmentally driven maternal effects and those associated to genetic variation. Authors state that by performing experiments with females from different seasons (summer vs autumn) they quantify variations in performance associated to genetic differences or prezygotic effects (L 129-132). So, if I understand well, they interpret the different responses found between individuals of the same season (intra-cohort) as maternal effects, and the differences between seasons (inter-cohort) as genetic differences and/or prezygotic effects. To my understanding, this is questionable because genetic variation is present both at the intra and inter-cohort level as long as you are comparing larvae from different females. This point needs to be clarified.

Answer: We have clarified this point in the “Introduction” and in the “Material and methods”. Both within and among cohort variation (which should reflect genetic variation + prezygotic effects) are separated from those studied here: the post-zygotic (environmental) maternal effects (driven by the temperature and salinity conditions experienced during embryogenesis). In the design, within cohort-variation is represented by the factor female (F) and between-cohort variation is represented by the factor season (S). The post-zygotic maternal effects are represented by the factors “embryonic salinity” (E_S) and “embryonic temperature” (E_T). Within cohort variation refers to variation in responses of individuals originated from different females but that experienced similar environments as embryos and larvae (this includes, same E_S , E_T , L_T , and L_S). Post-zygotic maternal effects are detected when offspring produced by different females, sharing the same condition during embryogenesis, have a different average response than offspring produced by females kept under a second environmental condition as embryos. In the design, a post-zygotic maternal effect is significant when the variation associated to such effect is larger than that associated to within-cohort variation. Because we studied two different cohorts, our design enabled to determine if the post-zygotic maternal effect differed among cohorts (as e.g. an interaction $S:E_T$).

We now make this clearer in the new version of the introduction (L120-126) by stating “...we performed the experiments with larvae from females producing eggs at different times of the year (autumn vs early summer), in order to determine if post-zygotic responses are consistent or if they vary among cohorts. Differences in responses among larvae from different cohorts (but otherwise kept under similar temperature-salinity conditions over both the embryonic and larval phases) should be driven by genetic differences among broods or the influence of prezygotic maternal effects²³.”

We have also clarified the definition of within- and between-cohorts effects in the “Material and methods” section by providing further information on the experimental design (L200-210):

1. “There was an additional factor, female (F) which represents the within-cohort variation in the responses i.e. variation in responses of individuals originating from different females and experiencing the same environments as embryos and larvae.”
2. “The between-cohort effect is captured by the term (S) and represents differences in the responses among individuals originating from females belonging to different cohorts and experiencing the same environments as embryos and larvae.”

Our general approach is similar to that used in other studies of maternal effects or plasticity in non-clonal organisms (e.g. Molluscs: Tanner et al. 2020; fish: Donelson et al. 2011, polychaetes: Massamba-N'Siala et al. 2014, cited in the paper), taking larvae produced by different females as being larvae from different families.

2. By exploring maternal effects from one generation to the next, potential pre-existing transgenerational effects are not controlled for. This is an unavoidable limitation because multiple generation studies are only feasible with species with short generation times, but it should be discussed.

Answer: We have clarified this point in the Discussion (L371-374) adding a reference.

Minor issues

Authors may consider the model of evolution of maternal effects proposed by Kuijper and Hoyle, 2015 *Evolution* 69-4: 950–968, on the evolutionary dynamics of maternal effects in highly variable or periodically fluctuating environments, to contextualize and interpret their results.

Answer: Kuijper & Hoyle (2015) show that adaptive maternal effects can evolve in the presence of sudden shifts in environmental conditions or when environmental change occurring at the scale of climate change (~100 years). We have added such information in the introduction (L82-85).

L 24 replace has with have

Done (L24).

L 26 replace invertebrate with invertebrates

Done (L26).

L 95-99 What physiological mechanisms underlie the mitigation effect of high temperature on low salinities? If there is information on this, please include it.

Answer: we have added a sentence describing the most likely mechanism based on available literature, which is covered in the Discussion (L101-103).

L 141-143 Were the females directly transferred to the treatments after field collection? Or were they allowed to habituate to laboratory conditions for some time before the experiments? If so, in which conditions were they maintained?

Answer: We added the following sentence: “On the day of collection, embryos were staged and females were distributed in the experimental treatment.” (L133-135).

L 148-149 Why did you measure carapace width? Was it a covariate considered in your analyses?

Answer: Carapace width is a standard measure of crab body mass which is not confounded by retention of seawater within the branchial chambers. We measured female

carapace to offer background information on the females that were used in the experiments. We added a sentence to explain that body size of the females had no effect: “We run preliminary correlation analyses with female size as a covariate, but we did not find any relationship. Therefore, we did not consider female body size in the subsequent analyses”. (L142-145).

L 161-163 Could you explain why the rearing conditions of the larvae are slightly different from that of the maternal environment? (i.e. different salinity in the low salinity treatment and one more temperature treatment for larvae).

Answer: We added a sentence to explain our rationale for using different conditions in the maternal and offspring treatments, as females and embryos have different tolerances to salinity. In addition, we wanted to test how larvae perform at higher temperatures by adding a treatment (24°C) to reflect warming under a climate change scenario (L159-162).

L 168-178 In this section, authors should be clearer in explaining which procedures were used for each salinity treatment. For example, in line 168 “supply with seawater” refers only to the higher salinity treatment, but not to the lower salinity one, and the same in L 172-174. But from L 175, I guess authors refer to the low salinity treatment.

Answer: We clarified the means by which seawater was diluted to obtain the low salinities (L166-178).

L 181-182 From each female, there were enough larvae to distribute them across 6 treatment combinations x 5 replicates x 10 individuals? Including information on the typical number of eggs laid by females of the study species would be useful.

Answer: We added this information and the fecundity information for *Carcinus maenas* in L183-184 (Young & Elliot 2020).

L 214-215 This part may not be clear for a reader non familiar with this analyses, authors should consider providing additional details.

Answer: We clarified that the singular matrix occurs when the method cannot estimate some of the components of the statistical model and we refer to Bolker et al. 2009 (L221-223).

L 363 In the context of this paper, it would be recommendable to specify that with “acclimatory responses” here you refer to a form of transgenerational or developmental plasticity, to distinguish it from within-generational acclimation responses.

Answer: We clarified that this is a form of developmental plasticity (L337-340).

Referee: 2

Comments to the Author(s)

I have read the manuscript entitled "Maternal and cohort effects modulate offspring responses to multiple stressors" by Torres et al. which describes how post-zygotic maternal effects modulate offspring performance in the invasive crab *C. maenas* in response to salinity and temperature. Overall, I found the paper well-written and the discussion of observed embryonic responses well-presented and therefore have only a few, minor, grammatical and visual changes to suggest in my comments (see below).

Line 24: Either “Current concerns ... have” or “Current concern ... has”

Done (L24)

Line 32: brooding instead of breeding?

Answer: Brooding is the correct terminology. Changed (L32)

Lines 37-39: Whereas the authors are correct that the number of studies of transgenerational plasticity (TGP) in brooders is small, the field is not entirely unexplored. See for example,

Tanner RL, Bowie RCK, Stillman JH. 2020. Thermal exposure and transgenerational plasticity influence embryonic success in a bivoltine estuarine sea hare. *Mar Ecol Prog Ser* 634:199-211. <https://doi.org/10.3354/meps13207>

Cumbo VR. et al. 2013. Brooded coral larvae differ in their response to high temperature and elevated pCO₂ depending on the day of release. *Mar Biol* 160, 2903–2917 (2013). <https://doi.org/10.1007/s00227-013-2280-y>

I therefore suggest the authors modify this sentence to note that their results support the data presented in these (and potentially other) studies. In addition, the authors should include discussion of this (and other relevant) literature in their paper where appropriate (e.g., in the Discussion).

Answer: We changed the emphasis of the last sentence of the Abstract (L37-39), which in the new version highlights that this study is a contribution towards a better understand how climate change may affect brooding organisms. As requested by the reviewer, we added the references (L388-390, L416-418, L432-435) in the context of the discussion of cohort effects, as both papers highlight the importance of phenotypic variation among cohorts if larvae are produced at different lunar cycles and years (Cumbo et al 2013) or different seasons (Tanner et al, 2020).

Lines 48-49: missing parentheses surrounding e.g. clause. Suggest re-wording of "...changes in offspring performance in individuals" as it is oddly phrased.

Answer: We reworded the phrase and added the missing parentheses (L48-50).

Figure 1 first panel should read "M = Optimal" and "S = Suboptimal". Suggest combining the "Pre-emption" and "Induction" panels as the optimal condition (TMLS) data is currently being displayed twice. This would also help clarify that, under optimal conditions, the TMLS response is exhibited whereas, sub-optimal conditions can lead to one of two alternatives (pre-emption and induction).

Answer: We changed the figure and the figure legend to highlight the possible responses. Note that the modified Figure 1 and the revised figure legend is in the new file "Figures and figure legends"

Line 342: Change “were” to “where”

Done (L315).

Line 365: Change “found” to “find”

Done (L340).

Appendix B

RESPONSE TO REFEREES

We thank again the reviewers for their new comments. Please find below detailed comments on each issue and information on the changes made in the manuscript.

Associate Editor:

Board Member: 1

Comments to Author:

Thanks for your patience with the review process. We have now got feedback on the revisions from both original reviewers of your manuscript. Both were very positive about the careful revisions you provided, but both (particularly reviewer 2) had some minor issues they suggested you address before the manuscript can be accepted. Personally I have nothing else to add. Congrats on a great contribution to the literature. The world needs more work on the complexity of multiple stressor interactions, so keep it up!

Answer: Many thanks for the encouraging words. We have addressed all the queries from the referees. Please see detailed comments below.

Reviewer(s)' Comments to Author:

Referee: 1

Comments to the Author(s)

I am happy to see that authors have satisfactorily addressed and clarified all the issues pointed in the previous revision round. I congratulate them for the considerable improvement of the manuscript. I have no further concerns on this new version.

I have detected these typos:

L 89: Replace assess with assessed

Answer: Done (L 89)

L 448 (word document) or 376 (pdf file): “and genetic variation” is repeated

Answer: Done (L 398)

Referee: 2

Comments to the Author(s)

The authors have adequately addressed my previous comments/suggestions and the revised manuscript is much improved. However, after careful revision, I feel that the Discussion is still slightly under-developed especially with regard to how this study fits within the context of transgenerational effects on population dynamics. It is currently unclear how variance in the observed physiological responses might affect population-level trajectories for this particular species. The authors do an excellent job of presenting possible effects of variation in general, either as a result of transgenerational or because of pre-zygotic effects, but more specific discussion of how this variation may affect populations of the study species *C. maenas* is needed.

Answer: Done, see comment to Line 364 (Discussion L 373-387).

In addition, there are a few other, minor, grammatical issues that I list below that should be corrected before publication. I hope my comments are useful to improve the manuscript.

Line 25: Change to “climate-driven”.

Answer: Done (L 25).

Line 25: Suggest to change “in particular on” to “particularly the”.

Answer: Done (L 25).

Lines 29 – 33: Suggest simplifying the sentence to “Here, we study how performance of offspring of a marine invertebrate (shore crab *Carcinus maenas*) changes in response to two

stressors (temperature and salinity) experienced by brooding mothers during embryogenesis in different seasons.”

Answer: Done (L 29-32).

Lines 43 – 44: Please include hyphens between words when they are used as a compound adjective (e.g., “multiple-driver (or stressor) effects”).

Answer: Done (L 43).

Line 81 – 82: Change to “(i.e., effects of the maternal environment or phenotype on 82 offspring phenotype and performance)”.

Answer: We added the comma after i.e. (L 81).

Line 118: Suggest to change to “...experienced by embryos during brooding on the survival...”.

Answer: Done (L 118).

Line 143: Change to “We ran ...”.

Answer: Done (L 146).

Line 184: Check value for fecundity. Is it 180 or 180,000? Units?

Answer: We added the units: embryos per clutch; 180000 is correct for the size of the females we used for this study, it is even higher for bigger females (L 187-188).

Line 186: Remove “*”.

Answer: Done (L 189).

Line 341: Remove “that” after “by contrast,”.

Answer: Done (L 350).

Line 352: "... with, for instance, the concept ..."

Answer: Done (L 361-362).

Line 364: The authors should expand their discussion of how their data fit within the context of "stabilizing or destabilizing the population dynamics". Does the inter-cohort variance in this study have the potential to affect population dynamics of *C. maenas* at the sites of collection? If so, in what way(s)?

Answer: Done. We added the following text: "For *C. maenas*, we do not have sufficient information about the structure of population and thus we can only speculate on how inter-cohort variation may affect the dynamics. Higher performance in larvae resulting from autumn embryos may contribute to recruitment by buffering offspring from potential suboptimal (winter-spring) conditions; this would be similar to a case where a few individuals of high quality secure resources and avoid a population crash⁶⁰. Larvae resulting from autumn embryos may also have a disproportionate contribution to recruitment (as compared with larvae from the spring-summer cohorts) if conditions are optimal. Disproportionate contributions to the population-biomass by embryos produced in winter (as compared to "summer embryos") appear to occur in natural populations of another coastal crustacean (brown shrimp *Crangon crangon*) in the North Sea⁶¹. Such embryos hatch into larvae that have higher tolerance to food limitation than those of summer embryos⁶², but the contribution of winter embryos to the population appears to occur through an additional number of factors (e.g. a seasonal pattern of mortality rate)" (L 373-387).